# Prognostic Value of Albumin to Globulin Ratio in Non-Metastatic and Metastatic Prostate Cancer Patients: A Meta-Analysis and Systematic Review

**DOI:** 10.3390/ijms231911501

**Published:** 2022-09-29

**Authors:** Stefano Salciccia, Marco Frisenda, Giulio Bevilacqua, Pietro Viscuso, Paolo Casale, Ettore De Berardinis, Giovanni Battista Di Pierro, Susanna Cattarino, Gloria Giorgino, Davide Rosati, Francesco Del Giudice, Alessandro Sciarra, Gianna Mariotti, Alessandro Gentilucci

**Affiliations:** 1Department of Maternal-Infant and Urologic Sciences, ‘Sapienza’ University of Rome, Policlinico Umberto I Hospital, 00100 Rome, Italy; 2Department of Urology, Humanitas, 20089 Milan, Italy

**Keywords:** prostatic neoplasm, albumin to globulin ratio, meta-analysis, radical prostatectomy, hormone therapy

## Abstract

The aim of our meta-analysis is to analyze data available in the literature regarding a possible prognostic value of the albumin to globulin ratio (AGR) in prostate cancer (PC) patients. We distinguished our analysis in terms of PC staging, histologic aggressiveness, and risk of progression after treatments. A literature search process was performed (“prostatic cancer”, “albumin”, “globulin”, “albumin to globulin ratio”) following the PRISMA guidelines. In our meta-analysis, the pooled Event Rate (ER) estimate for each group of interest was calculated using a random effect model. Cases were distinguished in Low and High AGR groups based on an optimal cut-off value defined at ROC analysis. Four clinical trials were enclosed (sample size range from 214 to 6041 cases). The pooled Risk Difference for a non-organ confined PC between High AGR and Low AGR cases was −0.05 (95%CI: −0.12–0.01) with a very low rate of heterogeneity (I^2^ < 0.15%; *p* = 0.43) among studies (test of group differences *p* = 0.21). In non-metastatic PC cases, the pooled Risk Difference for biochemical progression (BCP) between High AGR and Low AGR cases was −0.05 (95%CI: −0.12–0.01) (I^2^ = 0.01%; *p* = 0.69) (test of group differences *p* = 0.12). In metastatic PC cases, AGR showed an independent significant (*p* < 0.01) predictive value either in terms of progression free survival (PFS) (Odds Ratio (OR): 0.642 (0.430–0.957)) or cancer specific survival (CSS) (OR: 0.412 (0.259–0.654)). Our meta-analysis showed homogeneous results supporting no significant predictive values for AGR in terms of staging, grading and biochemical progression in non-metastatic PC.

## 1. Introduction

The management of prostate cancer (PC) and clinical outcomes after treatments is significantly influenced by PC heterogeneity. Clinical decisions continue to depend upon serum prostate specific antigen (PSA) levels, tumor stage, risk classes and pathological Gleason score [1,2]. PSA assay-based screening is affected by laboratory variability and low predictive value. Current clinical practice guidelines for early detection of prostate cancer recommend a personalized PSA-based management to improve the risk–benefit ratio of the screening strategy [3]. Moreover, it has to be considered that PSA value also increases in conditions of glandular inflammation that is consistent either in the population with benign hyperplasia or in that with PC, and it is often associated with the risk of performing unnecessary biopsies [4]. Predictive nomograms, mainly including these clinical parameters, are also used to evaluate the risk of advanced stage, undifferentiated tumors and progression after treatments [5,6]. 

Different data sustain the hypothesis that chronic inflammation plays a role in carcinogenesis and tumor progression. Several inflammatory mediators such as cytokines, chemokines and prostaglandins have been proposed as potential biomarkers for PC [7,8]. 

Hypoalbuminemia can be associated to systemic inflammation in patients with cancer [9]. Inflammatory reaction and immunity are influenced by serum albumin and globulin; hypoalbuminemia [10] and hyperglobulinemia are considered indicators of chronic inflammation in oncologic patients [11]. Albumin can reflect the body’s nutritional status and globulin the immunological and inflammatory status, and their ratio can be evaluated as albumin divided by total protein minus albumin value in serum [12]. Hypoalbuminemia was also studied in relation to fibrinogen values in other neoplastic diseases, such as in muscle-invasive bladder tumors. Authors showed that a low ratio was associated with poor differentiation, non-organ confined disease and independently predicted time to progression [13].

The serum albumin/globulin ratio (AGR) has been suggested as a prognostic marker for colorectal cancer, lung cancer, breast cancer and nasopharyngeal carcinoma [14,15,16,17].

A meta-analysis suggested that a low preoperative serum albumin to globulin ratio (AGR) is related to worse prognosis in different human neoplasms [12]. Some retrospective analyses described a prognostic value for AGR in PC patients, as well [18,19,20,21].

## 2. Methods

### 2.1. Evidence Acquisition

#### 2.1.1. Objective

The primary aim of this systemic review and meta-analysis is to analyze data available in the literature regarding a possible prognostic value of the albumin to globulin ratio (AGR) in PC patients. In particular, we distinguished our analysis either in terms of PC staging, histologic aggressiveness and risk of progression after treatments.

#### 2.1.2. Search Strategy

A literature search using electronic databases, such as PubMed, Medline, Web of Science, Scopus and the Cochrane library, of papers published in the last 20 years, was performed. The search process was performed on a combination of the items (“prostatic cancer” and “albumin” and “globulin” and/or “albumin to globulin ratio”) without language restrictions and following the Preferred Reporting Items for Systematic review and Meta-Analyses (PRISMA) guidelines. Original and review articles were included and critically considered. We have not included abstracts or reports from meetings.

#### 2.1.3. Selection of the Studies and Inclusion Criteria

Entry into the analysis was restricted to data collected from original studies on clinical retrospective or prospective trials including patients with a histological diagnosis of prostatic adenocarcinoma. Two authors (MF, GG) independently screened titles and abstracts of all articles using predefined inclusion criteria. The full-text articles were independently examined by three authors (MF, GG, GB) to determine whether they met the inclusion criteria. Then, two authors (MF, GG) extracted data from the selected articles. Final inclusion was determined by discussion of all investigators’ evaluation.

Studies selected for inclusion met the following criteria: (I) patients with a histological diagnosis of PC; (II) serum albumin to globulin ratio determination.

Articles were excluded if: (I) multiple reports were published on the same population; (II) data provided were insufficient for the outcomes described in the aim section; (III) failed to meet inclusion criteria; (IV) mixed populations without possibility of data extraction.

#### 2.1.4. Statistical Analysis

Risk of bias was assessed at the study level for each of the cohorts included, in full agreement with the Cochrane Collaboration’s “Risk of Bias” tool (Appendix A). According to predetermined endpoints, we compared the available populations using Standardized Mean Difference (SMD), Event Rate (ER) and Risk Difference (RD), with a 95% confidence interval (CI). An evaluation for the presence of heterogeneity was conducted using: (1) Cochran’s *Q*-test with *p* < 0.05 signifying heterogeneity; (2) Higgins I^2^ test with inconsistency index.

The pooled SMD, ER and RD estimate for each group was calculated using a random effects model, and our results are graphically displayed as forest plots. 

The possible prognostic value for AGR was estimated regarding PC staging, histologic aggressiveness and risk of progression after treatment. Calculations were accomplished using Stata version 1.7 (Stata Corporation, College Station, TX, USA) with all tests being two sided, and statistical significance set at <0.05.

### 2.2. Evidence Synthesis

#### 2.2.1. Studies Included in the Meta-Analysis

Database searches initially yielded 115 article references. Of these, 53 were subsequently removed due to either duplication or failure to meet the inclusion criteria. Full-text articles were then re-evaluated and critically analyzed for the remaining 62 references. Of these, 58 did not meet the inclusion criteria. The remaining four articles were considered for our critical review and meta-analysis (Figure 1 and Table 1 and Table 2).

#### 2.2.2. Quality of Studies and Sample Size

Of the four articles selected for the review [18,19,20,21], all studies were retrospective mono or multicenter clinical trials (Table 1).

The sample size of the populations ranged from 214 to 6041 cases across the 4 studies. All these studies defined the patient population in terms of clinical (age, PSA values), pathologic characteristics (staging, histologic grading) and progression (mainly biochemical (BCP)) after treatments. Two retrospective analyses [19,20] included non-metastatic PC cases considered for radical prostatectomy (RP), one [18] metastatic PC submitted to androgen deprivation therapies (ADT) and one [21] non-metastatic PC submitted to salvage RP after radiation therapy (Table 1).

#### 2.2.3. Identification of the Optimal Cut-Off for AGR

All four retrospective analyses [18,19,20,21] identified an optimal cut-off value for AGR so as to stratify the population between Low and High AGR groups. The AGR cut-off value was determined by receiver operating characteristics curve analysis, and the optimal cut-off in the different courts was similar, ranging from 1.31 to 1.53. Fewer cases in the Low versus High AGR groups were reported in all trials (Table 1).

**Table 1 ijms-23-11501-t001:** Four retrospective clinical trials included in the analysis and main characteristics of the trials and populations. AGR = albumin/globulin ratio. ADT = androgen deprivation therapy. Number of cases and percentage; mean ± SD; median (range).

Author	Year	Study Type	Population	AGR Groups	Total N of Patients	Age	PSA	TNM Stage	ISUP Grading	Follow-Up (Months)
**Chung JW et al. [20]**	2021	Retrospective	Non-metastatic PC submitted to radical prostatectomy	Low < 1.53 High ≥ 1.53	Tot: 742 Low AGR: 398 High AGR: 344	66.8 ± 6.3 66.8 ± 6.1	11.5 ± 11.0 12.7 ± 17.0	Low. pT2: 300 (75.4%) ≥pT3: 98 (24.6%) pN0: 382 (96%) pN1: 16 (4%) High. pT2: 297 (86.3%) ≥pT3: 47 (13.7%) pN0: 327 (95.1%) pN1: 17 (4.9%)	Low. 1–3: 313 (78.6%) 4–5: 85 (21.4%) High. 1–3: 294 (85.5%) 4–5: 50 (14.5%)	Low. 57.3 (48.2–72.7) High. 57.2 (41.9–74.8)
**Quhal F et al. [21]**	2021	Retrospective	Non-metastatic PC. Salvage radical prostatectomy after radiation therapy	Low < 1.4 High ≥ 1.4	Tot: 214 Low AGR: 89 High AGR: 125	69.0 (64–72)	5.8 (3.6–9.9) 5.3 (3.5–8.1) 6.9 (3.8–9.6)	Low. pT1: 33 (44%) pT2: 28 (37.3%) ≥pT3: 14 (18.7%) High. pT1: 47 (45.2%) pT2: 42 (40.4%) ≥pT3: 15 (14.4%)	Low. 1: 32 (43.8%) 2–3: 31 (42.5%) 4–5: 10 (13.7%) High. 1: 59 (56.2%) 2–3: 3 (31.4%) 4–5: 13 (12.4%)	25.3 (15–28.5)
**Aydh A et al. [19]**	2021	Retrospective	Non-metastatic PC submitted to radical prostatectomy	Low < 1.31 High ≥ 1.31	Tot: 6041 Low AGR: 2003 High AGR: 4038	61.0 (57–66)	6.0 (4–9)	Low. pT2: 1541 (76.9%) pT3a: 336 (16.8%) ≥pT3b: 126 (6.3%) High. pT2: 3133 (77.6%) pT3a: 670 (16.6%) ≥pT3b: 235 (5.8%)	Low. 1: 650 (32.5%) 2: 716 (35.7%) 3: 490 (24.5%) 4: 69 (3.44%) 5: 78 (3.89%) High. 1: 1282 (31.7%) 2: 1471 (36.4%) 3: 1022 (25.3%) 4: 133 (3.29%) 5: 130 (3.22%)	45.0 (35–58)
**Wang N et al. [18]**	2019	Retrospective	Metastatic PC submitted to ADT	Low < 1.45 High ≥ 1.45	Tot: 214 Low AGR: 100 High AGR: 114	70.7 ± 7.5 71.9 ± 7.6 69.6 ± 7.3	>20: 86.4% >20: 91.0% >20: 82.5%	M+	Low. 1: 12 (12%) 2–3: 37 (37%) 4–5: 51 (51%) High. 1: 12 (10.5%) 2–3: 52 (45.6%) 4–5: 50 (43.9%)	132

**Figure 1 ijms-23-11501-f001:**
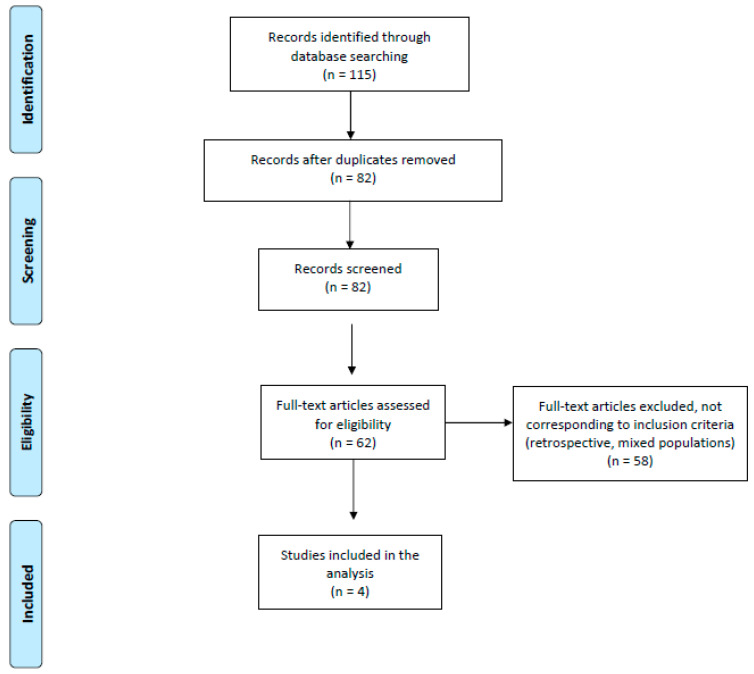
Flow chart for meta-analysis (PRISMA) [22].

## 3. Results and Discussion

### 3.1. Association between AGR and Pathologic Features

In three [19,20,21] out of the four studies (one study [18] included only metastatic PC) considering non-metastatic PC, cases showed a higher percentage of organ-confined than non-organ confined PC and a low percentage of lymph node involvement (LNI), without significant differences between Low and High AGR groups, although pT2 and pN0 cases were always more represented in High than in Low AGR groups (Table 2). A meta-analysis was implemented to examine the distribution of cases according to pathologic staging between High AGR and Low AGR groups. Considering a random effect model among eligible studies, the pooled Risk Difference for non-organ confined PC between High AGR and Low AGR cases was −0.05 (95%CI: −0.12–0.01) and that for lymph node involvement was 0.00 (95%CI: −0.02–0.02), with a very low rate of heterogeneity (I^2^ < 0.15%; *p* > 0.40) among studies (test of group differences *p* = 0.21) (Figure 2).

Deeks’ funnel plots are displayed in Appendix A, and meta-regression plots are presented in Appendix A. 

Similarly, in all studies [18,19,20,21], PC cases showed a higher percentage of ISUP grading 1–3 than ISUP grading 4–5, without significant differences regarding AGR groups, although ISUP 4–5 were always more represented in Low than in High AGR groups (Table 2). Considering a random effect model among eligible studies for our meta-analysis, the pooled Risk Difference for ISUP 4–5 PC between High AGR and Low AGR cases was −0.03 (95%CI: −0.07–0.02) with a very low rate of heterogeneity (I^2^ = 0.02%; *p* = 0.71) among studies (test of group differences *p* = 0.12) (Figure 2).

Deeks’ funnel plots are displayed in Appendix A, and meta-regression plots are presented in Appendix A. 

### 3.2. Association between AGR and Progression after Treatment

Results in terms of biochemical progression rates (BCP) were reported in the three analyses on non-metastatic PC submitted to RP [19,20,21]. Low AGR PC cases showed a higher incidence of BCP during postoperative follow-up, although differences did not reach statistical significance (Table 2). A meta-analysis was implemented to examine the distribution of cases according to BCP between High AGR and Low AGR groups. Considering a random effect model among eligible studies, the pooled Risk Difference for BCP between High AGR and Low AGR cases was −0.05 (95%CI: −0.12–0.01), with a very low rate of heterogeneity (I^2^ = 0.01%; *p* = 0.69) among studies (test of group differences *p* = 0.12) (Figure 2).

**Table 2 ijms-23-11501-t002:** Four retrospective clinical trials included in the analysis and results on the basis of AGR groups. AGR = albumin/globulin ratio. BCP = biochemical progression. B/C = Biochemical or Clinical progression. PFS = progression free survival. CSS = Cancer Specific Survival. Number of cases and percentage; mean ± SD; median (range); OR with 95% CI at multivariate analysis.

Author	AGR Groups	% of Cases on the Basis of Stage	AGR OR—95% CI Non-Organ Confined Disease	% of Cases on the Basis of ISUP	AGR OR—95%CI ISUP 4–5	% of Cases on the Basis of N	AGR OR 95%CI N+	% of Cases on the Basis of Progression after Treatment	AGR OR—95%CI Progression after Treatment	% of Cases Died for PC	AGR OR—95% CI for CSS
**Chung JW et al. [20]**	Low < 1.53 High ≥ 1.53	Low: pT2: 75.4% ≥pT3: 24.6% High: pT2: 86.3% ≥pT3: 13.7%	2.162 (1.430–3.269)	Low: 1–3: 78.6% 4–5: 21.4%. High: 1–3: 85.5 % 4–5: 14.5%	1.795 (1.171–2.752)	Low. N0: 96.0%; N+: 4.0% High. N0: 95.1% N+: 4.9%		Low: BCP-: 75.6% BCP+: 24.4% High BCP-: 79.1% BCP+: 20.9%	BCP 1.262 (0.898–1.773)		
**Quhal F et al. [21]**	Low < 1.4 High ≥ 1.4	Low T1: 44% T2: 37.3% ≥T3: 18.7% High T1: 45.2% T2: 40.4% ≥T3: 14.4%	1.22 (0.67–2.21)	Low. 1:43.8% 2–3: 42.5% 4–5: 13.7% High. 1: 56.2% 2–3: 31.4% 4–5: 12.4%		Low. N0: 79.8% N+: 20.2% High. N-: 82.4% N+ 17.6%	5.04 (1.69–15.03)	Low: BCP-: 51.7% BCP+: 48.3% High: BCP-: 62.4% BCP+: 37.6%	BCP 1.50 (0.96–2.39)		
**Aydh A et al. [19]**	Low < 1.31 High ≥ 1.31	Low ≤pT2: 76.9% pT3a: 16.8% ≥pT3b: 6.3 % High ≤pT2: 77.6% pT3a: 16.6% ≥pT3b: 5.8 %	1.01 (0.88–1.17)	Low. 1: 32.5% 2: 35.7% 3: 24.5% 4: 3.44% 5: 3.89% High 1: 31.7% 2: 36.4% 3: 25.3% 4: 3.29% 5: 3.22%		Low. N0: 99.5% N1: 0.5% High. N0: 99.2% N1: 0.8%	0.71 (0.46–1.12)	Low BCP-: 86.1% BCP+: 13.9% High BCP-: 90.1% BCP+: 9.9%	BCP 1.58 (1.36–1.83)		
**Wang N et al. [18]**	Low < 1.45 High ≥ 1.45			Low. 1: 12% 2–3: 37% 4–5: 51% High. 1: 10.5% 2–3: 45.6% 4–5: 43.9%				Low B/C- 32.0% B/C+:68.0% High B/C- 49.1% B/C+ 50.9%	PFS 0.642 (0.430–0.957)	Low Alive: 23.0% Died: 77.0% High Alive: 72.8% Died: 27.2%	0.412 (0.259–0.654)

Deeks’ funnel plots are displayed in Appendix A, and meta-regression plots are presented in Appendix A.

**Figure 2 ijms-23-11501-f002:**
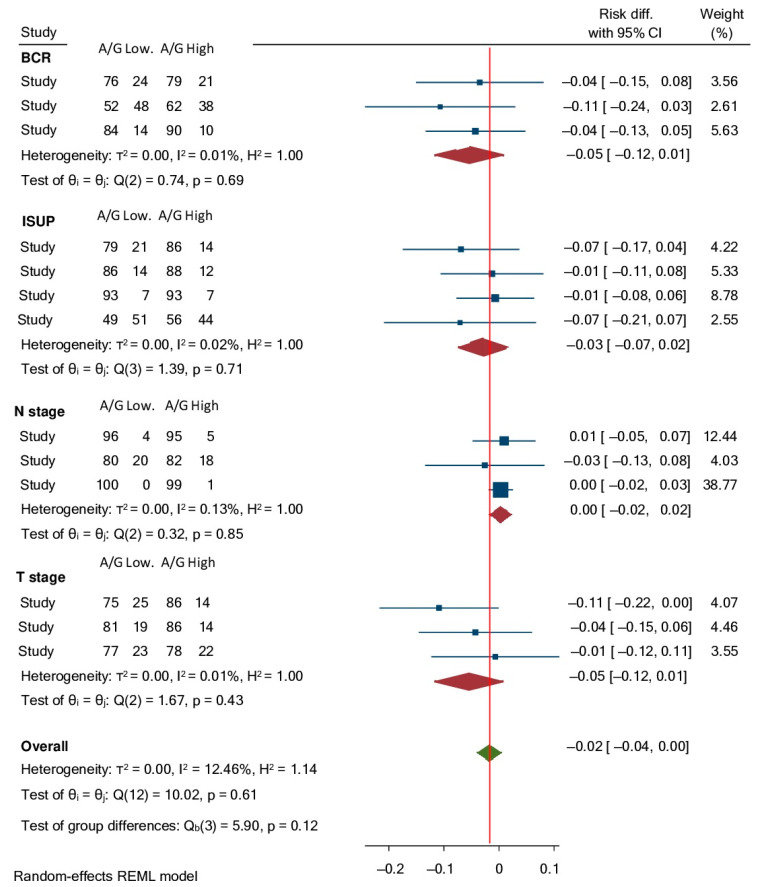
Forrest plot assessing Risk Differences with 95% CI for: biochemical recurrence (BCR), non-organ confined PC (T stage), lymph node involvement (N stage) and ISUP grading 4–5 between High and Low AGR groups.

Wang N et al. [18] analyzed metastatic PC cases submitted to ADT in terms of progression free survival (PFS defined as either biochemical or clinical progression) and cancer specific survival (CSS). Authors showed a significantly (*p* = 0.004) higher PFS rate in High AGR than in Low AGR cases with 68.0% of patients in the Low AGR and 50.9% in the High AGR group who experienced tumor progression. Moreover, in terms of CSS, a significantly (*p* < 0.05) higher percentage of cases in the Low AGR (77.0%) than in the High AGR (27.2%) died from PC.

### 3.3. Multivariate Analysis

At multivariate analysis, in non-metastatic PC cases submitted to RP and adjusted for clinical and pathological variables, AGR did not maintain an independent and significant predictive value in terms of BCP risk after treatment with OR ranging from 1.26 to 1.58 (*p* > 0.05) (Table 2).

On the contrary, in the only study [18] considering metastatic PC submitted to ADT, AGR showed an independent significant (*p* < 0.01) predictive value either in terms of progression free survival (PFS) (OR: 0.642 (0.430–0.957)) or cancer specific survival (CSS) (OR: 0.412 (0.259–0.654)).

### 3.4. Discussion

To our knowledge, this is the first meta-analysis evaluating the predictive value of albumin to globulin ratio (AGR) in terms of staging, histologic aggressiveness and progression risk after treatments in PC cases. In the present meta-analysis, following the PRISMA statements, we found only four retrospective analyses corresponding to our inclusion criteria. The quality of data from these four trials was limited by the retrospective analysis. Sample sizes ranging from 214 to 6041 cases were significant, and all these trials accurately defined the patient population in terms of pre-operative characteristics, pathologic results and progression after treatments. In particular, three out of the four selected analyses [19,20,21] considered non-metastatic PC cases selected for surgery (RP). All four studies identified an optimal cut-off value for AGR so as to stratify results between Low and High AGR groups. The AGR cut-off value was determined by receiver operating characteristics curve analysis, and the optimal cut-off in the different courts was similar, ranging from 1.31 and 1.53.

Our meta-analysis found a very low level of heterogeneity (I^2^ = 7.0%) of results among studies. In non-metastatic PC cases, dichotomizing results in Low and High, pretreatment AGR was not able to show a significant predictive value either in terms of pathologic features (T and N staging, ISUP grading) or in terms of biochemical progression risk. Considering a random effect model, the pooled Risk Difference for non-organ confined PC, lymph-node involvement and BCP between Low and High AGR groups was close to 0.00. Only one study [18] analyzed AGR in metastatic PC submitted to ADT. In this population, significant results were obtained either in terms of PFS or CSS prediction with a maintained independent (*p* < 0.01) value for AGR at multivariate analysis. Authors [18] showed 68.0% of patients in the Low AGR and 50.9% in the High AGR group who experienced tumor progression, and a higher percentage of cases in the Low AGR (77.0%) than in the High AGR (27.2%) group died from PC.

## 4. Conclusions

In a limited number of studies and with a retrospective design, we analyzed the prognostic value for albumin to globulin ratio in terms of pathologic staging, histologic aggressiveness or in terms of progression risk after treatments in PC cases. Our meta-analysis showed homogeneous results without significant differences in terms of AGR on the basis of PC staging, grading and biochemical progression. A potential prognostic role for AGR in non-metastatic PC cases seems to not be supported by the actual evidence.

## Data Availability

All data are available on electronic datasets PubMed, Medline, Web of Science, Scopus and the Cochrane library.

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
