# Peer review of "Prognostic Value of Albumin to Globulin Ratio in Non-Metastatic and Metastatic Prostate Cancer Patients: A Meta-Analysis and Systematic Review"

_ijms, 2022, doi:10.3390/ijms231911501_

Round 1

Reviewer 1 Report

This paper reports about the evidence that AGR has no prognostic value in the framework of PC. This information excludes the use of this ratio in the evaluation of the risk of the patient. Some main concepts should be implemented in the introduction of the paper. The sentence reported by the authors about the use of PSA tmay be prone to some criticisms

Lines 37-39. Authors should considee that there has been a change in the guidelines, quite recently, and that current clinical practice guidelines  for early detection of prostate cancer recommend for clinical decision-making a personalized prostate-specific antigen (PSA)-based management to improve the risk-benefit ratio of the screening strategy. Reference[Ferraro S, et al. Serum prostate specific antigen (PSA) testing for early detection of prostate cancer: Managing the gap between clinical and laboratory practice. Clin Chem. 2021;67:602-9.]

Furthermore when authors consider inflammation , lines 42-45, I think they should remind that the histological evidence of glandular inflammation is consistent in the population of patients with PCa, causing elevation of PSA results and thus  PSA results have to be properly managed.[reference:  Ferraro S,  et al. Definition of Outcome-Based Prostate-Specific Antigen (PSA) Thresholds for Advanced Prostate Cancer Risk Prediction. Cancers (Basel). 2021;13:3381-95.]

I will not encourage further evaluation of this ratio in metastatic PC population.

Author Response

ANSWERS TO REVIEWER’S CRITICISMS

We thank Reviewers for the positive comment and constructive requests. We answered to all their requests so to improve the quality of our analysis

Reviewer 1

This paper reports about the evidence that AGR has no prognostic value in the framework of PC. This information excludes the use of this ratio in the evaluation of the risk of the patient. Some main concepts should be implemented in the introduction of the paper. The sentence reported by the authors about the use of PSA tmay be prone to some criticisms

Q1: Lines 37-39. Authors should consider that there has been a change in the guidelines, quite recently, and that current clinical practice guidelines  for early detection of prostate cancer recommend for clinical decision-making a personalized prostate-specific antigen (PSA)-based management to improve the risk-benefit ratio of the screening strategy. Reference [Ferraro S, et al. Serum prostate specific antigen (PSA) testing for early detection of prostate cancer: Managing the gap between clinical and laboratory practice. Clin Chem. 2021;67:602-9.]

A1: As requested, in Introduction section, a brief description of personalized PSA-based management has been enclosed. The reference “Ferraro S, et al. Serum prostate specific antigen (PSA) testing for early detection of prostate cancer: Managing the gap between clinical and laboratory practice. Clin Chem. 2021;67:602-9.” was added.

Q2: Furthermore when authors consider inflammation, lines 42-45, I think they should remind that the histological evidence of glandular inflammation is consistent in the population of patients with PCa, causing elevation of PSA results and thus  PSA results have to be properly managed.[reference:  Ferraro S,  et al. Definition of Outcome-Based Prostate-Specific Antigen (PSA) Thresholds for Advanced Prostate Cancer Risk Prediction. Cancers (Basel). 2021;13:3381-95.]

A2: As requested, in introduction section, we had cited the PSA elevation in prostatic inflammation and we had cited the reference “Ferraro S,  et al. Definition of Outcome-Based Prostate-Specific Antigen (PSA) Thresholds for Advanced Prostate Cancer Risk Prediction. Cancers (Basel). 2021;13:3381-95.” .

Q3: I will not encourage further evaluation of this ratio in metastatic PC population.

A3: We had removed the last sentence of the Conclusion section about the necessity of further evaluation in metastatic PC population.

Reviewer 2 Report

While mentioning in the Introduction (line 46-56) the role of albumin in many human malignancies, I suggest to briefly cite the role of albumin in muscle invasive bladder cancer. EAU guidelines highlight the role of this protein in MIBC patients ' prognosis. Moreover some Authors explored the prognostic value of albumin in this setting with interesting findings (doi: 10.1016/j.urolonc.2021.04.026). The Authors should consider to concisely discuss these results.

Author Response

ANSWERS TO REVIEWER’S CRITICISMS

We thank Reviewers for the positive comment and constructive requests. We answered to all their requests so to improve the quality of our analysis

Reviewer 2

Q1: While mentioning in the Introduction (line 46-56) the role of albumin in many human malignancies, I suggest to briefly cite the role of albumin in muscle invasive bladder cancer. EAU guidelines highlight the role of this protein in MIBC patients ' prognosis. Moreover some Authors explored the prognostic value of albumin in this setting with interesting findings (doi: 10.1016/j.urolonc.2021.04.026). The Authors should consider to concisely discuss these results.

A1: As requested, in introduction section, we had briefly described the role of the Albumine/fibrinogen ratio in MIBC and we cited the work “Claps F, Rai S, Mir MC, van Rhijn BWG, Mazzon G, Davis LE, Valadon CL, Silvestri T, Rizzo M, Ankem M, Liguori G, Celia A, Trombetta C, Pavan N. Prognostic value of preoperative albumin-to-fibrinogen ratio (AFR) in patients with bladder cancer treated with radical cystectomy. Urol Oncol. 2021 Dec;39(12):835.e9-835.e17. doi: 10.1016/j.urolonc.2021.04.026. Epub 2021 May 26. PMID: 34049782.”

Round 2

Reviewer 1 Report

No further comments.